The zebrafish as a model system for analyzing mammalian and native α-crystallin promoter function

Posner Mason mposner@ashland.edu 1
Murray Kelly L. 1
McDonald Matthew S. 1
Eighinger Hayden 1
Andrew Brandon 1
Drossman Amy 1
Haley Zachary 1
Nussbaum Justin 2
David Larry L. 3
Lampi Kirsten J. 4
1 Department of Biology/Toxicology, Ashland University , Ashland , OH , United States of America
2 Department of Biology, Lakeland Community College , Kirtland , OH , United States of America
3 Department of Biochemistry and Molecular Biology, Oregon Health and Science University , Portland , OR , United States of America
4 Department of Integrative Biosciences, Oregon Health and Science University , Portland , OR , United States of America
Petrie Kevin
Electronic publication date: 2017 Nov 27
Publication date: 2017
Volume: 5
Electronic Location ID: e4093
Received 2017 Mar 20; Accepted 2017 Nov 4
Copyright: ©2017 Posner et al.
Copyright year: 2017
Copyright holder: Posner et al.
License: This is an open access article distributed under the terms of the Creative Commons Attribution License, which permits unrestricted use, distribution, reproduction and adaptation in any medium and for any purpose provided that it is properly attributed. For attribution, the original author(s), title, publication source (PeerJ) and either DOI or URL of the article must be cited.
License URL: https://creativecommons.org/licenses/by/4.0/

Keywords: Zebrafish, Lens, Crystallins, Promoters, GFP, Proteomics, Mass spectrometry, Gene expression, Vision

Funding: National Eye Institute of the National Institutes of Health EY027012 EY10572 This work was supported by an R15 AREA grant from the National Eye Institute of the National Institutes of Health to Mason Posner (EY013535) and from grants to support faculty/student research from the Provost Office of Ashland University. A summer student research stipend was provided to Kelly L. Murray as part of a Choose Ohio First scholarship grant to Ashland University. The proteomic analysis was supported by National Eye Institute grants EY027012 and EY10572 to Larry David and Kirsten Lampi. There was no additional external funding received for this study.

==============================
Previous studies have used the zebrafish to investigate the biology of lens crystallin proteins and their roles in development and disease. However, little is known about zebrafish α-crystallin promoter function, how it compares to that of mammals, or whether mammalian α-crystallin promoter activity can be assessed using zebrafish embryos. We injected a variety of α-crystallin promoter fragments from each species combined with the coding sequence for green fluorescent protein (GFP) into zebrafish zygotes to determine the resulting spatiotemporal expression patterns in the developing embryo. We also measured mRNA levels and protein abundance for all three zebrafish α-crystallins. Our data showed that mouse and zebrafish αA-crystallin promoters generated similar GFP expression in the lens, but with earlier onset when using mouse promoters. Expression was also found in notochord and skeletal muscle in a smaller percentage of embryos. Mouse αB-crystallin promoter fragments drove GFP expression primarily in zebrafish skeletal muscle, with less common expression in notochord, lens, heart and in extraocular regions of the eye. A short fragment containing only a lens-specific enhancer region increased lens and notochord GFP expression while decreasing muscle expression, suggesting that the influence of mouse promoter control regions carries over into zebrafish embryos. The two paralogous zebrafish αB-crystallin promoters produced subtly different expression profiles, with the aBa promoter driving expression equally in notochord and skeletal muscle while the αBb promoter resulted primarily in skeletal muscle expression. Messenger RNA for zebrafish αA increased between 1 and 2 days post fertilization (dpf), αBa increased between 4 and 5 dpf, but αBb remained at baseline levels through 5 dpf. Parallel reaction monitoring (PRM) mass spectrometry was used to detect αA, aBa, and αBb peptides in digests of zebrafish embryos. In whole embryos, αA-crystallin was first detected by 2 dpf, peaked in abundance by 4–5 dpf, and was localized to the eye. αBa was detected in whole embryo at nearly constant levels from 1–6 dpf, was also localized primarily to the eye, and its abundance in extraocular tissues decreased from 4–7 dpf. In contrast, due to its low abundance, no αBb protein could be detected in whole embryo, or dissected eye and extraocular tissues. Our results show that mammalian α-crystallin promoters can be efficiently screened in zebrafish embryos and that their controlling regions are well conserved. An ontogenetic shift in zebrafish aBa-crystallin promoter activity provides an interesting system for examining the evolution and control of tissue specificity. Future studies that combine these promoter based approaches with the expanding ability to engineer the zebrafish genome via techniques such as CRISPR/Cas9 will allow the manipulation of protein expression to test hypotheses about lens crystallin function and its relation to lens biology and disease.

Introduction

The zebrafish has become a valuable model system for examining lens development, disease and the function of lens crystallin proteins. Multiple studies have identified genes and proteins involved in lens formation (Yang et al., 2004; Yang & Cvekl, 2005; Vihtelic, 2008) and taken advantage of zebrafish embryo transparency to produce detailed imagery of lens development (Greiling & Clark, 2009). Patterns of lens development are similar between zebrafish and mammals, with a prominent exception being that in mammals the lens placode invaginates into the lens vesicle, while in zebrafish lens fiber cells delaminate from the placode (Greiling, Aose & Clark, 2010). Changes in the zebrafish lens proteome during its development have been described (Greiling, Houck & Clark, 2009; Greiling, Aose & Clark, 2010; Wages et al., 2013). The crystallin protein content of the zebrafish lens has been detailed (Posner, Kantorow & Horwitz, 1999; Runkle et al., 2002; Wistow et al., 2005; Smith et al., 2006; Posner et al., 2008), and functional studies have examined zebrafish α-crystallin chaperone-like activity and stability in comparison to mammals (Dahlman et al., 2005; Koteiche et al., 2015). Multiple ocular diseases, such as glaucoma, diabetic retinopathy, macular degeneration and cataract have been modeled in the zebrafish (Morris, 2011; Gestri, Link & Neuhauss, 2012; Chhetri, Jacobson & Gueven, 2014). In total these studies illustrate the benefits of using the zebrafish to study lens biology and provide insights into the normal function and dysfunction of the vertebrate lens.

One area of zebrafish lens biology that has not been well explored is the activity and function of lens crystallin promoters. Kurita et al. (2003) cloned the zebrafish αA-crystallin promoter region and used it to drive the expression of diphtheria toxin in the lens to study developmental connections between lens and retina. Davidson et al. (2003) used a Xenopus γ-crystallin promoter to express green fluorescent protein (GFP) in the zebrafish lens. Goishi et al. (2006) constructed a zebrafish αA-crystallin promoter/GFP plasmid to show how the zebrafish cloche mutant, which lacks a functional DNA-binding transcription factor implicated in vascular development (Reischauer et al., 2016), might downregulate αA-crystallin expression. To our knowledge, no work since these studies has utilized zebrafish crystallin promoters, and no study has characterized the temporal or spatial expression of reporter genes linked to these promoters.

The function of mammalian α-crystallin promoters has been the subject of multiple studies. Examination of the shared promoter region between αB-crystallin and HspB2 in the mouse identified specific regions that enhance αB-crystallin expression. For example, an enhancer spanning −426∕ −259 was required for extralenticular expression while a more proximal region from −164∕ + 44 produced reporter gene expression in lens (Dubin et al., 1991; Gopal-Srivastava, Kays & Piatigorsky, 2000; Swamynathan & Piatigorsky, 2002). To recapitulate endogenous expression of mouse αB-crystallin, four kilobases of the 5′-flanking promoter sequence was needed (Haynes, Duncan & Piatigorsky, 1996). A region spanning −111 to +46 of the mouse αA-crystallin promoter was shown to drive expression of GFP in both cultured lens cells and in the mouse, with this expression enhanced by inclusion of a distal enhancer approximately 8 kilobases upstream of the gene (Yang & Cvekl, 2005; Yang et al., 2006). While no published studies report the use of mouse lens crystallin promoters in the zebrafish, Hou et al. (2006) showed that a fragment of the human βB1-crystallin promoter produced transgenic expression of GFP in the zebrafish lens. A subsequent study used this human promoter to drive the expression of novel proteins in the zebrafish lens to examine the function of aquaporin water channels (Clemens et al., 2013). The evolutionary conservation of lens crystallin gene regulation is not surprising considering the similar expression of lens crystallin proteins between zebrafish and mammals (Posner et al., 2008; Greiling, Houck & Clark, 2009). This conservation suggests that mammalian α-crystallin promoters could be functionally assessed in the zebrafish, providing a faster and less expensive system than traditional mouse transgenic approaches. The growing development of zebrafish gene editing techniques would greatly expand the capabilities of this system. Data on crystallin promoter activity would also facilitate the expression of introduced proteins in zebrafish lens and other tissues.

A comparison of mouse and zebrafish α-crystallin promoter activity can also help detail the evolution of tissue specific expression. Past studies have examined the evolution of crystallin gene expression at different timescales. For example, sequence comparisons have detailed the recruitment of crystallins during the initial evolution of the vertebrate lens, finding that the α-crystallins are related to extra-lenticular small heat shock proteins (Wistow & Piatigorsky, 1988). A subsequent gene duplication event was followed by divergence in transcriptional regulation and expression between the two resulting paralogs (αA and αB-crystallin) (Cvekl et al., 2017). A more recent evolutionary change in the regulation of α-crystallins was investigated in the blind mole rat, in which the αB-crystallin promoter has specifically lost lens activity, presumably reflecting the degenerated eyes of this subterranean species (Hough et al., 2002). In this present study we further examine evolutionary changes in α-crystallin expression by comparing promoter activity of the two divergently expressed zebrafish αB-crystallin paralogs. While the expression of these proteins is already known in adults, an examination of their gene’s promoter activities and protein abundance during early development can identify possible ontogenetic shifts in expression. The structure, stability, chaperone-like activity and expression pattern of zebrafish αBb-crystallin is similar to the mouse ortholog (Dahlman et al., 2005; Smith et al., 2006). We predicted that this conservation would extend into early development. However, it is an open question whether the altered expression of the lens-specific zebrafish αBa-crystallin begins in early development, or appears later in ontogeny.

Our results suggest that the zebrafish can be used as a time efficient and cost effective model for screening the activity of mammalian lens crystallin promoters. Comparison between orthologous mouse and zebrafish promoter activity supports the hypothesis that α-crystallin promoter function is conserved between these species. Our comparative promoter analysis of the two zebrafish α-crystallins shows a subtle difference in expression, and timing of developmental upregulation, between these two paralogs. We also show that zebrafish αBa-crystallin undergoes an ontogenetic shift in its expression to become lens-specific later in development.

Materials and Methods

Zebrafish maintenance and breeding

AB or ZDR strain zebrafish were housed in 10 L aquaria on a recirculating filtering system maintained at 28−30 °C with a 14:10 h light and dark cycle. Fish were fed twice each day with either commercial flake food or live Artemia brine shrimp. Two males and two females were placed in one liter breeding tanks the afternoon prior to morning egg collections. Plastic dividers were used to separate the two sexes until eggs were needed to assure that all embryos were of similar ages. All animal procedures were approved by Ashland University’s Animal Care Committee (approval number MP 2015-1).

Table 1 Primers used to construct promoter fragments.

Gene accession numbers are given for each gene as well as coordinates showing region of each promoter fragment relative to the gene’s start codon. Lower case letters indicate added nucleotides for insertion into cloning plasmid. Promoter fragments were constructed through either traditional cloning methods using restriction enzymes Xho1 and BamH1 or by Gibson assembly. The mouse αA-crystallin promoter was provided by the laboratory of Dr. Ales Cvekl.

Gene	Gene accession #	Coordinates	Forward primer	Reverse primer	Construction method	
Zebrafish	
αA 1 kb	NM_152950.2	−1028∕ −1	ggaattctcgagTGGAGACCCCTGATTAATA	ggaattggatccAATGTCAGACCTGGTAACT	Xho1/BamH1	
αBa 3 kb	NM_131157.1	−3000∕ −1	ctaccggactcagatcGAAAAAAAAA- AAAGAAAGAAAGAAAAGAAAG	ccatggtggcgaccggtgTGTACCTTAGTTTGGAGC	Gibson	
αBb 1 kb	NM_001002670.2	−1074∕ −1	ggaattctcgagTTCAATGGTGCGCTGT	ggaattggatccTTTGAGTCTGGGCCTCTT	Xho1/BamH1	
αBb 2 kb	−2092∕ −1	ggaattctcgagAGACGTTACAGTGGGCTA	ggaattggatccTTTGAGTCTGGGCCTCTT	Xho1/BamH1	
αBb 4 kb	−3999∕ −1	ctaccggactcagatcCGCACCGTACAAAGATTTG	ccatggtggcgaccggtgTTTGAGTCTGGGCCTCTTC	Gibson	
αBb 5 kb	−4999∕ −1	ctaccggactcagatcAATTTAGACCTGCTTTTAGTTGG	ccatggtggcgaccggtgTTTGAGTCTGGGCCTCTTC	Gibson	
Mouse	
αA	NM_001278570	−7706∕ − 7492 and −1800∕ + 46		
αB 0.25 kb	CT010341	−259∕ −1	cgagctcaagcttcgGTGAAACAAGACCATGAC	ccatggtggcgaccggtgTGTGGCTAGATGAATGCAG	Gibson	
αB 0.8 kb	−833∕ −1	ggaattctcgagGTGCAGCTATGAGGGTGTGA	ggaattggatccTGTGGCTAGATGAATGCAGA	Xho1/BamH1	
αB 1.5 kb	−1501∕ −1	ggaattctcgagAAAGCAAGAGGCAGGATGAG	ggaattggatccTGTGGCTAGATGAATGCAGA	Xho1/BamH1	

Comparative analysis of α-crystallin promoter regions

The UCSC Genome Browser (http://genome.ucsc.edu/; Kent et al., 2002) was used to identify conserved regions in the mouse and zebrafish αA-and αB-crystallin promoters (Fig. S1). A previous analysis of syntenic relations was used to assess the rearrangement of gene relationships after duplication of zebrafish αB-crystallin (Elicker & Hutson, 2007).

Promoter expression plasmid construction, embryo injection and assessment of GFP expression

Primers used to amplify regions of each α-crystallin promoter were designed using DNA Main Workbench based on sequences in GenBank and ordered from Sigma Genosys (Table 1). Each promoter region was then amplified from a bacterial artificial chromosome (BAC) clone obtained from the BACPAC Resources Center (bacpac.chori.org) using Platinum Pfx DNA polymerase (Thermo Fisher; Waltham, MA, USA). Amplification conditions were optimized to produce single bands of the expected size, which were then subcloned into the pJET1.2 plasmid (Thermo Fisher) and sequenced to confirm their identity (Functional Biosciences, Madison, WI, USA). Restriction enzyme sites designed into each amplification primer were used to digest and ligate each cloned promoter into the pAcGFP1-1 plasmid (Clontech, Mountain View, CA, USA) using enzymes from New England Biolabs (Ipswich, MA). NEB 5 alpha cells (NEB) were transformed with each promotor/GFP construct. Some promoter constructs were produced using an alternate Gibson Assembly approach using the company’s protocol (NEB). All cloned promoters have been deposited with Addgene (http://www.addgene.org/Mason_Posner/). A promoter/GFP construct for mouse αA-crystallin was provided by Dr. Ales Cvekl.

To prepare promoter expression plasmids for injection into zebrafish embryos, plasmids were linearized with NotI, purified using a QIAquick PCR Purification kit (Qiagen, Valencia, CA, USA) and then dialyzed with TE buffer using a 0.025 µm VSWP membrane (Millipore, Billerica, MA, USA). Injection solutions contained 35 ng/ul of the dialyzed plasmids, 0.2% phenol red and a sufficient volume of 0.1 M KCl to produce 5 microliters of injection mix. Two nanoliters of this solution was injected into one-cell stage zebrafish embyros with a Harvard Apparatus PL-90 picoinjector (Holliston, MA, USA) using needles prepared with a Sutter P97 Micropipette Puller (Novato, CA, USA). Injection pressures were adjusted to inject 1 nl of plasmid solution with each 20 ms pulse. Injected embryos and uninjected controls were incubated at 28 °C in fish system water and transferred to 0.2 mM PTU at 24–30 h post fertilization to block melanin production and facilitate observation of GFP expression.

The presence of any GFP expression was examined using an Olympus IX71 inverted microscope and imaged with a SPOT RT3 camera (Diagnostic Instruments, Sterling Heights, MI, USA). Live embryos were anesthetized in tricaine and imaged at 100× or 200× total magnification using UV illumination and GFP filter. Confocal images were captured on a Leica SP5 microscope after embryos were anesthetized and fixed in 4% paraformaldehyde. Embryos for confocal imaging were mounted on slides using Vectashield (Vector Laboratories, Burlingame, CA, USA). Image series were then rendered as three-dimensional surface projections using Volocity imaging software (Perkin Elmer, Waltham, MA, USA).

Quantitative PCR analysis of α-crystallin expression in embryos

All qPCR reactions were conducted in the investigators’ laboratory and were designed to meet MIQE guidelines when possible as described below (Bustin et al., 2009). ZDR strain zebrafish were placed in breeding tanks that separated males and females until tank dividers were removed. Resulting fertilized eggs were collected and incubated in petri dishes containing system water in a 28 °C incubator. Embryos were removed at 12 h post fertilization (hpf), 24 hpf, 2 days post fertilization (dpf) 3 dpf, 4 dpf and 5 dpf and chilled on ice before replacing system water with RNAlater (Thermo Fisher) and then stored in a −20 °C freezer until RNA purification. Embryos were stored between 1 h and several days. Approximately 15 embryos were used from each timepoint for RNA purification, and three different sets of embryos were collected at each timepoint to produce three biological replicates. Total RNA from each sample was purified using an RNEasy Minikit (QIAGEN) with Qiashreddor and quantified with a NanoDrop 1000 Spectrophotometer (Thermo Scientific). Chorions were not removed from embryos that had not yet hatched. Purified total RNA (2,000 ng) from each sample was treated with DNaseI (NEB) and 6 µl was used to synthesize cDNA using the Protoscript II First Strand cDNA Synthesis Kit (NEB) with the oligo d(T)23 primer. The resulting cDNA sample was calculated to contain the equivalent of 16 ng/µl of original purified RNA.

All cDNA samples (three biological replicates for each timepoint) were amplified using Luna Universal qPCR Master Mix (NEB) on an Applied Biosystems StepOne Real-Time PCR System (Thermo Fisher). We used three endogenous control primer sets previously published in past studies (Tang et al., 2007; McCurley & Callard, 2008) and a primer pair for each of the three zebrafish alpha crystallins as designed by Elicker & Hutson (2007). All primers used and related information are shown in Table 2. Each reaction was performed in triplicate using cDNA equivalent to 32 ng of initial purified RNA and each primer at a final concentration of 250 nm in 20 µl reactions with the following parameters: hold at 95 °C for 1 min; 40 cycles of 95 °C for 15 s and 60 °C for 30 s; fast ramp setting.

Table 2 Primers used for qPCR analysis of zebrafish α crystallin expression.

All primers have been used in previous publications (see Methods for references). Standard curve qPCR reactions were used to calculate the efficiency and R2 value for each primer pair when amplifying cDNA from transcribed adult zebrafish lens RNA.

Gene	Primer sequence	Product size (bp)	Accession #	Efficiency	R2	
αA-crystallin	F: 5′ATGGCCTGCTCACTCTTTGT3′ R: 5′CCCACTCACACCTCCATACC3′	159	AY035778	84.3	0.965	
αBa-crystallin	F: 5′CCCAGGCTTCTTCCCTTATC3′ R: 5′GTGCTTCACATCCAGGTTGA3′	196	NM_131157	97.5	0.994	
αBb-crystallin	F: 5′CCTATCGACGGCAAATGTT3′ R: 5′GGCATCAGCAGCAGACAATA3′	128	NM_001002670	93.8	0.995	
EF-1α	F: 5′CAGCTGATCGTTGGAGTCAA3′ R: 5′TGTATGCGCTGACTTCCTTG3′	94	AY422992	85.4	0.999	
β-actin	F: 5′CGAGCAGGAGATGGGAACC3′ R: 5′CAACGGAAACGCTCATTGC3′	102	FJ915059	80.5	0.997	
Rpl13A	F: 5′TCTGGAGGACTGTAAGAGGTATGC3′ R: 5′AGACGCACAATCTTGAGAGCAG3′	148	NM_212784	96.1	0.999	

Melt curve analysis was used to confirm that a single product was produced (95 °C for 15 s, 60 °C for one minute). A set of qPCR products from each primer pair was electrophoretically analyzed on a gel as well to confirm that products were the appropriate size. Each qPCR product was also sequenced to confirm that primers had amplified the correct gene. Water was used as a non-template control to detect the presence of any contaminating DNA. Parallel RNA samples from every timepoint and biological replicate that had not been treated with reverse transcriptase were amplified in duplicate as a –RT control.

The Applied Biosystems StepOne software (version 2.1) was used to calculate Ct values for each reaction using the software’s default settings. All Ct values were exported and further calculations made in Excel (Supplemental Information 1). The Ct values for the three technical replicates from each endogenous control reaction were calculated. Forty-nine out of 51 sets of technical triplicates produced Ct values within 0.5 cycles. The averaged triplicate Ct values for each of the three endogenous control gene were then themselves averaged to produce an overall average for each sample. The Ct values produced by each of the three α-crystallin primer pairs was also calculated for every cDNA sample by averaging the values from each technical triplicate. A delta Ct was then calculated for every cDNA sample for each α-crystallin gene by subtracting the average endogenous control Ct for that sample from the values measured with the α-crystallin specific primer pair. The result was a delta Ct for each of three biological replicates for three α-crystallin genes at each timepoint. These values were imported into the statistical package R (R Core Team, 2017) (using R Studio (R Studio Team, 2015)) and visualized by box and whisker plots. R was also used to determine any statistically significant differences between timepoints for each alpha crystallin (ANOVA with post-hoc Tukey’s HSD test).

Standard curves were generated for each primer pair to measure efficiency percentage. Purified RNA from adult zebrafish lenses was used as template for these standard curves as alpha crystallin expression in embryos was too low to produce amplification across a large range of template concentrations. Lenses were surgically removed from anesthetized adult zebrafish and purified total RNA was DNaseI treated prior to cDNA synthesis as described above.

Proteomic analysis of α-crystallin content in zebrafish

A pair of lenses from adult zebrafish were dissected, placed in 100 µl of 50 mM ammonium bicarbonate buffer, and probe sonicated (3 × 5 s with cooling on ice between treatments) to produce a uniform suspension. The protein concentration was then determined using a BCA assay (Thermo Fisher) using BSA as a standard. A 50 µg portion of protein was then reduced, alkylated, and trypsinized in the presence of ProteaseMax™ detergent using the manufacturer’s recommended protocol (Promega, Madison, WI, USA). Following digestion, trifluoroacetic acid was added at a final 0.5% concentration, the sample centrifuged at 16,000 × g for 5 min, and the supernatant transferred to an autosampler vial. One μg of digest was then loaded onto an Acclaim PepMap 0.1 × 20 mm NanoViper C18 peptide trap (Thermo Fisher) for 5 min at a 5 µl/min flow rate in a 0.1% formic acid mobile phase. Peptides were then separated using a PepMap RSLC C18, 2 µm particle, 75 µm × 25 cm EasySpray column (Thermo Fisher) and 7.5–30% acetonitrile gradient over 60 min in mobile phase containing 0.1% formic acid at a 300 nl/min flow rate using a Dionex NCS-3500RS UltiMate RSLCnano UPLC system. Data-dependent tandem mass spectrometry data was collected using an Orbitrap Fusion Tribrid mass spectrometer configured with an EasySpray NanoSource (Thermo Fisher). Survey scans from 400–1,600 m∕z were performed in the Orbitrap mass analyzer at 120,000 resolution, automatic gain control (AGC) setting of 4.0 × 105, 50 ms maximum injection time, and lock mass using a m∕z = 445.12 polysiloxane ion. Data-dependent MS2 scans on peptide ions with signal intensities higher than 5,000, ranging from +2 to +6 charge state, and passing the monoisotopic precursor selection filter were selected for higher energy collision dissociation (HCD) with a 30% collision energy using quadrupole isolation with a 1.6 m∕z window. Fragment ions were then analyzed in the linear ion trap with an AGC setting of 1.0 × 104, maximum injection time (MIT) of 35 ms, dynamic exclusion enabled, repeat count of 1, exclusion duration of 30 sec, exclusion mass tolerance of ±10 ppm, top speed mode, and 3 s dwell time between Orbitrap survey scans. MS/MS results were then matched to peptide sequences using Sequest HT software within the Protein Discoverer 1.4 suite (Thermo Fisher) using a UniProt database containing the taxon identifier 7955 (Danio rerio) generated in July 2016 and containing 58,290 entries. Searches were performed with trypsin specificity, a maximum of 2 missed cleavages, precursor and fragment ion tolerances of 10 ppm and 1 Da, respectively for parent and daughter ions using monoistopic masses. A static modification of +57.02 Da was added to all cysteine residues due to alkylation with iodoacetamide, and a variable modification of +15.99 Da for methionine oxidation. Peptide identifications were filtered with the Percolator node in Protein Discoverer using a reverse sequence database strategy to estimate peptide false discovery. The resulting Protein Discoverer .msf file was then imported into Skyline software (version 3.6.0.10162) (MacLean et al., 2010) to create a spectral library using identified peptides with Percolator q scores ≤0.05, having between 8–25 residues, and no missed cleavages. Three peptides each for entries Q8UUZ6 (αA-crystallin), Q9PUR2 (αBa-crystallin), and Q6DG35 (αBb-crystallin) were selected based on manual observation of parent ion intensities and quality of fragment ion spectra. These were then used to create a parallel reaction monitoring method to detect the presence of the three α-crystallins during embryo development.

Figure 1 Confocal imagery showing representative sites of GFP expression produced by mouse and zebrafish α-crystallin promoters.

Examples of lens expression produced with a zebrafish αA promoter (A and B). Various sites of extraocular expression shown as single z-planes (on left) and as 3-dimensional renders (on right) for skeletal muscle produced with a mouse αB promoter (C and D); for notochord produced with a zebrafish αBb promoter (E and F); dorsal to the yolk produced with a zebrafish aA promoter (G and H).

Uniform suspensions of either whole embryos or dissected embryo eyes and trunks were created using either probe sonication in 50 mM ammonium bicarbonate as above, or by vortexing vigorously for 30 min in 20 µl of 50 mM ammonium bicarbonate buffer containing 0.2% ProteaseMax detergent. Following a protein assay, from 10–50 µg of each suspension was digested using the ProteaseMax protocol as recommended by the manufacturer, and 2 µg of each digest analyzed by LC/MS using the same chromatographic separation and instrument as above, except using a parallel reaction monitoring method (Bourmaud, Gallien & Domon, 2016) to detect the 9 targeted α-crystallin peptide ions (Table S1). Peptides were isolated and fragmented as above, except without data-dependency and by cycling through the list of ions throughout the chromatographic separation so the intensity of fragment ions could be continuously monitored. MS/MS spectra were acquired in the instrument’s Orbitrap mass analyzer at a resolution of 30,000, AGC setting of 5 × 104, 100 ms MIT, with a scan range of m∕z 200–2,000. Skyline was then used to extract intensities for the three most intense fragment ions for each peptide determined from the lens spectral library, and perform peak detection and integration to monitor the relative abundance of α-crystallins during embryo development.

Results

The location of green fluorescent protein (GFP) expression resulting from injection of mouse and zebrafish promoters into zebrafish zygotes was examined by both standard fluorescent microscopy and confocal microscopy. Representative confocal images of anatomical structures that expressed GFP during this study are shown in Fig. 1. Video fly-throughs of representative structures can be found in Videos S1–S3. Patterns and timelines of expression produced by each promoter can be found in Tables 3 and 4, and are described below. Overall 1,622 observations were made of 616 individual injected embryos ranging in age from 24 h post fertilization (hpf) to 7 days post fertilization (dpf). GFP expression was seen in 76.0% of examined embryos.

Table 3 Location of promoter activity.

Total embryos shows the number of separate embryos examined after injection with each indicated promoter fragment. Percentage of embryos shows the proportion of GFP-expressing embryos with observable GFP in each tissue. A “O” indicates no embryos expressed GFP in that tissue. “Eye” indicates expression other than the lens, “NC” indicates notochord, “SM” indicates skeletal muscle.

	Total embryos	Percentage of embryos	
		Lens	Eye	NC	SM	Heart	
Zebrafish promoters	
1 kb αA	62	97	O	5	11	O	
3 kb αBa	90	3	7	56	55	O	
1 kb αBb	67	O	5	5	90	O	
2 kb αBb	59	O	6	3	92	3	
4 kb αBb	51	O	O	20	100	O	
5 kb αBb	90	4	4	7	97	O	
Mouse promoters	
αA	55	70	O	4	30	O	
0.25 kb αB	64	27	2	62	79	2	
0.8 kb αB	50	13	15	33	94	6	
1.5 kb αB	67	15	5	42	100	12	

Table 4 Timeline of promoter activity.

Numbers indicate percentage of injected embryos expressing GFP in any tissue at indicated timepoints. Lack of expression is noted with an “O” and “–” indicates that no embryos were observed at that timepoint.

	Hours post fertilization (hpf)	
	24	30	48	54	72	78	
Zebrafish promoters	
1 kb αA	O	O	61	83	84	–	
3 kb αBa	O	O	90	40	52	61	
1 kb αBb	O	O	63	56	62	26	
2 kb αBb	O	O	67	–	47	47	
4 kb αBb	O	O	58	87	90	–	
5 kb αBb	O	O	100	–	83	50	
Mouse promoters	
αA	O	17	27	39	32	–	
0.25 kb αB	23	65	95	100	–	–	
0.8 kb αB	O	19	100	100	91	–	
1.4 kb αB	O	33	85	80	72	–	

Figure 2 Comparison of mouse and zebrafish αA-crystallin chromosomal arrangement and their ability to drive GFP expression in zebrafish embryos.

The structural and functional conservation of mammalian and zebrafish αA-crystallin is mirrored in their shared syntenic relationship with hsf2bp (A). Vertical bars note exons, thin horizontal lines note introns and arrows show direction of transcription. The promoter regions for each gene produced similar temporal and spatial expression patterns (B–E), with expression almost exclusively restricted to the lens. The extent of lens expression varied for both orthologous promoters (compare B to C for mouse and D to E for zebrafish).

Mouse and zebrafish αA-crystallin promoters produced similar GFP expression in zebrafish embryos with a subtle difference in timing

Previous work has shown strong conservation in αA-crystallin DNA sequences, protein stability and chaperone-like activity between zebrafish and mammals (Runkle et al., 2002; Dahlman et al., 2005; Posner et al., 2012). The zebrafish and mouse αA-crystallin orthologs are similarly arranged relative to other genes, with both located in a head-to-head orientation with heat shock factor binding protein gene hsf2bp (Fig. 2A). However, mouse and human contain a second gene, salt inducible kinase 1 between αA and hsf2bp, and the intergenic distances are much greater (Wolf et al., 2008). Several sequence regions of the mouse αA-crystallin promoter are conserved in the zebrafish genome (Fig. S1A). Here we show that a mouse αA-crystallin promoter fragment (−111 to +46) combined with enhancer region DCR1 drove green fluorescent protein expression in the zebrafish lens, with much less common expression in skeletal muscle (Figs. 2B–2C; Table 3). This pattern was similar to that produced by a 1 kb fragment of the zebrafish αA promoter (Figs. 2D–2E). The zebrafish promoter also produced spots of GFP expression dorsal to the yolk that were much less common with the mouse promoter (Figs. 1G–1H). A small fraction of embryos injected with the zebrafish and mouse αA-crystallin promoters showed GFP expression in segments of the notochord. There was a subtle difference in onset of expression between the two promoters, with GFP driven by the mouse αA promoter noticeable by 30 h post fertilization (hpf) while the zebrafish αA promoter became active between 30 and 48 hpf (Table 4).

Mouse αB-crystallin promoter drove GFP expression in zebrafish embryos

Previous studies have shown that an upstream enhancer region (−426/−257) of the mouse αB-crystallin promoter is required for extralenticular expression, while a more proximal region (−164∕ + 44) is sufficient for driving lens expression (Dubin et al., 1991; Gopal-Srivastava & Piatigorsky, 1994) (Fig. 3A). Genomic sequence alignment shows two areas of conservation between mouse and zebrafish in this proximal promoter region (Fig. S1B). We cloned three mouse αB-crystallin promoter fragments into a GFP plasmid to examine whether these functional regions had similar effect in zebrafish. Our results indicate that the mouse αB-crystallin promoter drives GFP expression in zebrafish embryo skeletal muscle, notochord, lens and heart (Fig. 3). The presence of the upstream enhancer region increased expression in skeletal muscle and heart compared to the shorter 250 bp fragment (Table 3). The 0.8 and 1.5 kb promoters both produced skeletal muscle GFP expression in a large percentage of embryos (94 and 100%), while this percentage was a lower 79% when using the 250 bp fragment. (Table 3). Heart GFP expression was reduced from 6% and 12% to 2% with the 250 bp promoter. However, exclusion of the upstream enhancer increased the number of embryos expressing GFP in the lens and notochord (Table 3). The 250 bp promoter fragment also led to slightly earlier GFP expression (by 24 h post fertilization) than the longer 800 bp and 1.5 kb fragments (Table 4).

Figure 3 Mouse αB-crystallin promoter fragments produced native expression in zebrafish embryos.

Enhancer elements of a promoter upstream of mouse αB-crystallin were previously shown to regulate expression in skeletal muscle (sm), heart and lens (lsr1 and 2) (A; adapted from Swamynathan & Piatigorsky, 2002). Fragments containing 250 bp, 0.8 and 1.5 kb lengths of this promoter produced GFP expression in zebrafish embryo notochord (B–D), skeletal muscle (E), lens (F) and heart (G; arrows). (E) shows GFP expression in both fast (noted by *) and slow twitch (noted by arrows) muscle fibers. The yolk remaining in these embryos is autofluorescent. The 250 bp fragment, which lacked the heart and skeletal muscle enhancer, produced less frequent GFP expression in these tissues, and GFP expression onset was slightly earlier (Tables 3 and 4).

Figure 4 The paralogous zebrafish αBa- and αBb-crystallin promoters produced similar, but distinct, GFP expression profiles.

Zebrafish αBb-crystallin has the same syntenic relationship with Hspb2 as mouse αB-crystallin, although the intergenic region between the two genes is much larger in the zebrafish (A). The zebrafish αBa-crystallin paralog has moved to a separate chromosome. Both zebrafish paralogs produced GFP expression most often in notochord (B) and skeletal muscle (C). The αBa paralog drove expression in these tissues equally while αBb was more active in skeletal muscle (D). Expression in lens (E) and extralenticular regions of the eye was more rare. Images shown are representative with the details of GFP expression not differing noticeably between paralogs or the promoter length used.

The two zebrafish αB-crystallins produced different patterns of GFP expression

The presence of two αB-crystallins in the zebrafish is likely the result of an ancient genome duplication event at the base of teleost evolution (Van de Peer, Taylor & Meyer, 2003). This duplication resulted in a divergence in chromosomal arrangement. Zebrafish αBa is located on chromosome 15 along with several distant genes with which its ortholog shows syntenic relationship in mammals (Elicker & Hutson, 2007). Zebrafish αBb has maintained the same tail-to-tail organization with fellow heat shock protein Hspb2 as is found in mammals, however the intergenic region in the zebrafish is much larger, at 6 kilobases compared to 1 kb in the mouse (Fig. 4A).

Previous studies indicated that the expression pattern of zebrafish αBa- and αBb-crystallin differs in adults, and proteomic analysis showed a difference in the timing of expression onset (Posner, Kantorow & Horwitz, 1999; Smith et al., 2006; Wages et al., 2013). However, no study has characterized developmental patterns produced by the two respective promoter regions. We produced a GFP-linked 3 kb fragment of the zebrafish αBa-crystallin promoter and a series of GFP-linked fragments spanning the expanded αBb-crystallin promoter. We found no difference between the timing of onset for any of these zebrafish promoters, with GFP first appearing between 30 and 48 hpf (Table 4). We also found no difference in timing or spatial expression between the αBb-crystallin promoter fragments, suggesting that sequences upstream of 1 kb do not regulate expression of this gene.

The spatial expression of GFP produced by all of the zebrafish αB-crystallin promoters was similar, with expression common in skeletal muscle and notochord (Figs. 4B–4C). However, the prevalence of GFP in these two tissues differed, with the zebrafish αBa promoter driving GFP equally (54.9% in skeletal muscle and 56.3% in notochord) while zebrafish αBb promoters were much more active in skeletal muscle than notochord (95.5% versus 10.3%; Fig. 4D). Both zebrafish αB promoters produced very rare GFP expression in lens (seven embryos out of 357 observed; Fig. 4E), some expression in the eye peripheral to the lens (22 embryos) and three αBb promoter-injected embryos, out of 267, produced GFP expression in the heart. Overall these data suggest that the divergent expression of the two zebrafish αB-crystallin promoters previously identified in adults appears later in development than the 1–7 dpf window examined in this present study.

Measurement of zebrafish α-crystallin transcription by qPCR

We used quantitative polymerase chain reaction (qPCR) to measure the concentration of α-crystallin mRNA during zebrafish embryo development. Messenger RNA from all three α-crystallins was detectable at 0.5 dpf, the earliest time point analyzed (Fig. 5). The variation in Ct values within each technical triplicate for this and other early timepoints was high for all three α-crystallins, suggesting that α-crystallin expression was very low at these early developmental stages. Transcription of αA-crystallin increased between 1 and 2 dpf, while αBa-crystallin expression increased between 4 and 5 dpf. Variation in Ct values within each technical triplicate decreased to less than 0.5 as expression increased. Transcription levels for αBb-crystallin remained consistently low through 5 dpf. Ct values for all three α-crystallins were much higher than those for the three reference genes, suggesting that expression of all α-crystallins was relatively low. All Ct values and delta Ct value calculations are shown in Supplemental Information 1.

Figure 5 qPCR analysis of α-crystallin expression in zebrafish embryos.

Box and whisker plot shows delta Ct for the three zebrafish α-crystallins, indicating mRNA levels relative to three endogenous controls from 12 h post fertilization (0.5 dpf) to 5 dpf. Lower numerical Ct values on these inverted y-axes indicate increased expression. All three graphs show low initial baseline expression that increases in αA (A) and αBa-crystallin (B), but stays consistently low in αBb-crystallin (C). Alpha A-crystallin expression increased earlier than αBa-crystallin expression. Asterisks indicate statistically significant differences in expression compared to the 0.5 dpf timepoint (p < 0.05; ANOVA with a post-hoc Tukey’s HSD test). Each box plot reflects three separate biological replicates for each timepoint. The calculated Ct values for each of three technical triplicates making up each biological replicate showed variation of more than 0.5 only for the lower expressed baseline samples. Timepoints with statistically significant increased expression produced technical triplicates with Ct values within 0.5 of each other (with one exception out of 9 measurements). Three and 4-dpf samples were not analyzed for αA- and αBa-crystallin since no change in expression was seen for each gene between 2 and 5 dpf.

Figure 6 Relative abundance of αA- and αBa-crystallin proteins in zebrafish embryos during development measured by mass spectrometric parallel reaction monitoring of tryptic peptides.

(A) Changes in αA-crystallin relative abundance in whole embryos from 1–6 days post fertilization (dpf) by measurement of peak areas for the top three fragment ions of peptide 52–65 (NILDSSNSGVSEVR). Orange, y12; blue, y11; and green, y10 fragment ions. The bar labeled library shows the relative proportion of these fragment ions for this peptide identified in a digest from an adult zebrafish lens, while the dotp value above each bar is a measurement of how well the observed fragment ions for this peptide in each embryo digest matched those for this peptide in a spectral library created from an adult lens digest. Note that the relative peak area for the library peptide was arbitrarily set to the same value as the largest peak area for ease of comparison. (B) Relative abundance of αA-crystallin in dissected eyes and remaining trunks of either 4 or 7 dpf embryos. The same αA peptide and fragment ions as measured above in A were used. (C) Measurement of αBa-crystallin in whole embryos from 1–6 dpf by measurement of peak areas for the top 3 fragment ions of peptide 79–88 (HFSPDELTVK). Orange, b2; blue, y9; and green, y8 fragment ions. (D) Relative abundance of αBa-crystallin in dissected eyes and remaining trunks of either 4 or 7 dpf embryos. The same αBa peptide and fragment ions as measured in C were used. Extracted ions chromatograms for the fragment ions of these peptides are shown in Fig. S2.

Proteomic analysis identified two of three α-crystallins in zebrafish embryos

We used a mass spectrometric parallel reaction monitoring approach to identify the presence of α-crystallins in pooled zebrafish embryos at 1 to 6 dpf as a complement to the promoter expression and qPCR data presented above. Two of the three targeted αA-crystallin peptides were detected by 2 dpf and peaked in abundance at 4–5 dpf. The results for αA peptide 52–65 from whole embryo digests are shown in Fig. 6A, while Fig. 6B shows the relative abundance of αA-crystallin in dissected eye and remaining trunks at 4 and 7 dpf. These results indicate that αA-crystallin peptide was largely present only in eye. Its apparent decrease in abundance in whole embryos by 6 dpf was likely due to its dilution by non-ocular proteins during embryonic development. While only one αBa-crystallin was detected in embryo digests, its measurement indicated that αBa-crystallin was present in almost equal abundance from 1–6 dpf (Fig. 6C). While small amounts of αBa-crystallin were detected in 4 dpf trunks, the protein was not detected in trunks by 7 dpf (Fig. 6D). However, αBa-crystallin was present in eye and appeared to increase from 4 to 7 dpf. The decrease in αBa in trunk and concurrent increase in eye is consistent with its unaltered abundance in whole embryos from 1–6 dpf. No αBb-crystallin peptides were detected in either whole embryos or dissected eyes or trunks. The extracted ion chromatograms from these parallel reaction monitoring experiments are shown in Figs. S2–S8.

Discussion

This present study is the first to show that mammalian α-crystallin promoter function can be analyzed in zebrafish embryos by observing green fluorescent protein (GFP) expression, suggesting that the zebrafish can be used as an efficient model for mammalian α-crystallin promoter analysis. We also provide the first data characterizing the activity of the three zebrafish α-crystallin gene promoters. These data detail their spatiotemporal expression and identify differences between embryonic and adult expression patterns for the duplicated and divergent zebrafish αB-crystallin paralogs. Future studies can use the techniques described here to measure the expression potential of modified lens crystallin promoters. Our data also show how different crystallin promoters could be used to drive the expression of target genes in specific tissues. Lastly, by examining promoter activity, mRNA expression and protein abundance for zebrafish α-crystallins, we resolve questions about the timing of expression of these genes.

Our data show that mouse α-crystallin promoters successfully drive expression in zebrafish embryos. An interesting question is whether the resulting expression patterns match that expected in the mouse, or alternatively, if the zebrafish embryos read the mouse promoter in their own way. Is mouse promoter activity modified by the signaling molecule environment of the zebrafish? This question is difficult to answer for the mouse αA-crystallin promoter as the GFP expression produced in zebrafish embryos was very similar to that of the native zebrafish ortholog. Similar expression profiles produced by each αA-crystallin promoter could be due to evolutionarily conserved roles in development, or alternatively that the zebrafish embryo reads the mouse promoter as one of its own. Interestingly, the mouse αA promoter expressed GFP at a slightly younger age than the native zebrafish promoter. This difference was also seen when comparing the mouse and zebrafish αB-crystallin promoters. Some element in the mouse promoter sequences appears to have accelerated the timing of expression. Earlier studies showed that the transcription factors Pax6, c-Maf, and CREB regulate expression of mouse αA-crystallin (Yang & Cvekl, 2005) and that FGF signaling regulates expression of c-Maf (Xie et al., 2016). We hypothesize that teleost fishes use a similar regulatory system, although the presence of two Pax6 genes in zebrafish may alter the details of this regulation (Kleinjan et al., 2008). Analysis of mouse and chicken βB1-crystallin promoter regions showed similar cross-species conservation in regulation, with some differences that indicated additional regulatory elements for lens-specific expression in the mouse promoter (Chen et al., 2001). The lack of detected αA peptides outside of the lens and low prevalence of extraocular GFP expression resulting from αA-promoters is consistent with the interpretation that the presence of αA-crystallin protein outside the lens is low throughout zebrafish development.

Comparison of mouse and zebrafish αB-crystallin promoter function is complicated, and potentially more interesting, because of the presence of two αB-crystallin paralogs in the zebrafish. Zebrafish αBa-crystallin protein is largely restricted to the lens in adults while αBb-crystallin is found ubiquitously, similar to the single mammalian ortholog (Posner, Kantorow & Horwitz, 1999; Smith et al., 2006). Since duplicated αB-crystallins are only known from teleost fishes such as the zebrafish, the restriction of expression to the lens likely evolved after the genome duplication event in this taxon (Van de Peer, Taylor & Meyer, 2003). There are several interesting observations to note about the comparisons between mouse and zebrafish orthologs and between the two zebrafish paralogs. First, the mouse αB-crystallin promoter drove lens GFP expression in a larger percentage of embryos than either zebrafish αB-crystallin promoter. This result may reflect the lower overall abundance of α-crystallin in the zebrafish lens compared to mammals (Posner et al., 2008), and previously observed low levels of αBa- and αBb-crystallin expression in early zebrafish development (Greiling, Houck & Clark, 2009; Wages et al., 2013). Second, the proportion of embryos expressing GFP in notochord and skeletal muscle varied between the three different promoters. The zebrafish αBb promoter was a strong driver of GFP expression in skeletal muscle, like the mouse αB promoter, but was not as active in the notochord. The zebrafish αBa promoter was active in notochord, like the mouse promoter, but less active in skeletal muscle. These observations are consistent with a hypothesis that mammalian αB-crystallin function has been divided between the two zebrafish paralogs. While the function of αB-crystallins in zebrafish notochord and skeletal muscle at these developmental stages is not known, it is possible that some developmental functions in these tissues have been divided between the two zebrafish paralogs as well. Signaling sequences from an original teleost αB-crystallin gene have possibly become split between the two current zebrafish paralogs. Finally, as mentioned above, the mouse αB-promoter initiated GFP expression at an earlier stage than the zebrafish orthologs.

The length of each zebrafish αBb-crystallin promoter fragment had no noticeable effect on GFP expression, suggesting that any regulatory elements influencing this gene’s activity remain within the first 1 kb upstream of the start codon. Regulatory elements do not appear to have been “stretched out” with the inclusion of additional sequence between zebrafish αBb-crystallin and Hspb2. We did, however, see a noticeable difference in expression when a mouse αB-crystallin promoter without the skeletal muscle and heart enhancer regions was used. This short 250 bp promoter fragment increased expression in lens and decreased expression in skeletal muscle. It also appeared to increase notochord expression. The significance of α-crystallin notochord expression is not known, although it was previously identified in mouse embryos (Gernold et al., 1993). These differences in expression produced with each mouse αB-crystallin promoter length are some of our best evidence that α-crystallin regulatory elements are conserved between mouse and zebrafish.

There are some conflicting results between studies that have examined α-crystallin expression in zebrafish embryos. A qPCR analysis by Elicker & Hutson (2007) detected αA- and αBb-crystallin mRNA starting at 12 hpf, with αA increasing steadily through 5 dpf and αBb increasing more slowly. They did not detect αBa-crystallin mRNA through 5 dpf. Another study using RT-PCR also found no αBa mRNA in zebrafish embryos through 78 hpf (Mao & Shelden, 2006). A recent report by Zou et al. (2015) found steady expression of αBb-crystallin between 24 hpf and 3 dpf, similar to Elicker and Hutson. However, their detection of αBa-crystallin expression by RT-PCR starting at 24 hpf with steady increase through 5 dpf differs from both past studies. Our data encompassing promoter driven GFP-expression, qPCR analysis, and proteomics help to address these discrepancies. Our detection of an increase in αA-crystallin mRNA at 2 dpf is consistent with the Elicker and Hutson study. This steady increase in expression was mirrored by a steady rise in αA-peptide detected in our embyos by mass spectrometry. While our approach does not allow us to directly compare amounts of mRNA or peptide between the three zebrafish α-crystallins, our data show that αA-crystallin expression increases first, suggesting that it is the most abundant of the three α-crystallins in early development. This conclusion is supported by a shotgun proteomic study from Greiling, Houck & Clark (2009) that found αA-crystallin, but no αBa- or αBb-crystallin, in 4.5 dpf zebrafish embryos. Our finding that αBa-crystallin mRNA levels increased between 4 and 5 dpf conflicts with Elicker and Hutson. Mao and Shelden only examined the 78 hpf timepoint and may have missed the increase. Our proteomics data identified similar levels of αBa-crystallin peptide between 1 and 6 dpf, suggesting that while mRNA levels are low, and increase at 5 dpf, protein levels remain consistent during this developmental period. We did not see the large increase in αBa-crystallin mRNA identified by Zou et al. between 24 hpf and 3 dpf. The lack of αBb-crystallin peptide in our proteomics data is consistent with the lack of increase in mRNA from our qPCR data. We did not see the steady increase in αBb-crystallin mRNA found by Elicker and Hutson, but rather the steady expression identified by Zou et al. However, the lack of detectable peptide, lack of increased mRNA (compared to increase in the other two zebrafish α-crystallins), and larger variance in Ct values for technical triplicates all suggest that αBb-crystallin is expressed at very low levels, if at all, through 5 dpf. Futhermore, the low abundance of GFP lens expression produced by both zebafish αB-crystallin promoters suggests that any significant protein expression that occurs through 5 dpf is outside of the lens. However, our mass spectrometry did identify more αBa peptide in the eye compared to the rest of the zebrafish body. At some, yet unidentified, point in zebrafish development αBa-crystallin expression becomes restricted to the lens.

In total the results of this study show that mammalian α-crystallin promoter function can be screened efficiently in zebrafish embryos. Controlling regions in these promoters appear to be well conserved. Comparison of the duplicated zebrafish αB-crystallin promoters provides insight into how the function of the single mouse αB-crystallin may have been divided between its two zebrafish orthologs. We also show that the lens specificity of zebrafish αBa-crystallin seen in adults does not occur in the embryo. Variation in the temporospatial expression produced by the ten promoter fragments analyzed in this study provide a new toolset for directing the expression of introduced proteins in various embryonic zebrafish tissues at different stages of development. Our combined analysis of zebrafish α-crystallin promoter activity, mRNA expression and protein abundance also clarifies discrepancies in the literature about when and where these genes are expressed. The ease with which engineered promoters can be injected into zebrafish embryos and their expression patterns visualized makes this model species ideal for analyses of protein expression regulation. Future studies that combine these promoter based approaches with the expanding ability to engineer the zebrafish genome via techniques such as CRISPR/Cas9 will allow the manipulation of protein expression to test hypotheses about lens crystallin function and its relation to lens biology and disease.

Supplemental Information

Video S1 Three dimensional render of GFP expression in zebrafish notochord produced by a zebrafish αBb-crystallin promoter fragment

Click here for additional data file.

Video S2 Three dimensional render of GFP expression in zebrafish skeletal muscle produced by a mouse αB-crystallin promoter fragment

Click here for additional data file.

Video S3 Three dimensional render of GFP expression dorsal to the yolk produced by a zebrafish αA-crystallin promoter fragment

Click here for additional data file.

Figure S1 Sequence alignments of upstream regions of mouse αA (A) and αB-crystallin (B) showing conservation across vertebrate taxa

Green boxes highlight conserved regions between fishes and mammals. Images include the full extent of zebrafish genomic sequence available in the alignment. The two conserved regions in the αB alignment may be due to retention of two lens specific enhancers in the mouse promoter. However, our data suggest that these lens enhancers are not functional through 7 days post fertilization.

Click here for additional data file.

Figure S2 Targeted mass spectrometric detection of αA-crystallin peptide 52–65 (NILDSSNSGVSEVR) in digests of 1–6 day post fertilization (dpf) embryos

Spectral library: MS2 spectrum of αA peptide 52–65 created during a data-dependent analysis of adult zebrafish lens digest. Y10-12 fragment ions were the most abundant in this spectrum, and these were used to detect the peptide in embryo digests, based on their simultaneous elution at approximately 29.5 min during the LC/MS analyses (colored traces marked with an arrow). These fragment ion peaks were integrated for each digest from 1–6 dpf embryos, and results are shown in the Peak Integration Results bar graph, indicating that the αA-crystallin reached its greatest abundance at 4 days dpf. The bar in the graph labeled Library shows the relative proportion of the y10-12 ions in the MS2 spectrum from the lens library, set at the same relative abundance as the fragment ions in 4 dpf digest. The relative intensities of the y10-12 fragment ions detected in the 2–6 dpf samples were very similar to those observed in the MS2 spectrum from the library, as evidenced by their dot product (dotp) values ranging from 0.97 to 0.99 marked above each bar.

Click here for additional data file.

Figure S3 Targeted mass spectrometric detection of αA-crystallin peptide 89–99 (VTDDYVEIQGK) in digests of 1–6 day post fertilization (dpf) embryos

Spectral library: MS2 spectrum of αA peptide 89–99 created during a data-dependent analysis of adult zebrafish lens digest. Y 5, 6, and 9 fragment ions were the most abundant in this spectrum, and these were used to detect the peptide in embryo digests, based on their simultaneous elution at approximately 25.8 min during the LC/MS analyses (colored traces). These fragment ion peaks were integrated for each digest from 1–6 dpf embryos, and results are shown in the Peak Integration Results bar graph, indicating that the αA-crystallin reached its greatest abundance at 5 days dpf. The bar in the graph labeled Library shows the relative proportion of the y 5, 6, and 9 ions in the MS2 spectrum from the lens library, set at the same relative abundance as the fragment ions in 5 dpf digest. The relative intensities of the y 5, 6, and 9 fragment ions detected in the 2–6 dpf samples were very similar to those observed in the MS2 spectrum from the library, as evidenced by their dot product (dotp) values ranging from 0.97 to 0.99 marked above each bar.

Click here for additional data file.

Figure S4 Targeted mass spectrometric detection of αBa-crystallin peptide 79–88 (HFSPDELTVK) in digests of 1–6 day post fertilization (dpf) embryos

Spectral library: MS2 spectrum of αBa peptide 79–88 created during a data-dependent analysis of adult zebrafish lens digest. b2, y8, and y9 fragment ions were the most abundant in this spectrum, and these were used to detect the peptide in embryo digests, based on their simultaneous elution at approximately 27.5 min during the LC/MS analyses (colored traces). These fragment ion peaks were integrated for each digest from 1–6 dpf embryos, and results are shown in the Peak Integration Results bar graph, indicating that the αBa-crystallin remained in nearly equal abundance during 1–6 dpf. The bar in the graph labeled Library shows the relative proportion of the b2, y8, and y9 ions in the MS2 spectrum from the lens library, set at the same relative abundance as the fragment ions in the 3 dpf digest. The relative intensities of the b2, y8, and y9 fragment ions detected in the 1–6 dpf samples were very similar to those observed in the MS2 spectrum from the library, as evidenced by their dot product (dotp) values ranging from 0.94 to 0.97 marked above each bar.

Click here for additional data file.

Figure S5 Targeted mass spectrometric detection of αA-crystallin peptide 52–65 (NILDSSNSGVSEVR) in digests of 4 and 7 day post fertilization (dpf) dissected embryo eyes and trunks

Spectral library: MS2 spectrum of αA peptide 52–65 created during a data-dependent analysis of adult zebrafish lens digest. Y10-12 fragment ions were the most abundant in this spectrum, and these were used to detect the peptide in embryo digests, based on their simultaneous elution at approximately 25.5 min during the LC/MS analyses (colored traces marked with an arrow). These fragment ion peaks were integrated for each digest from 4 and 7 dpf embryo eyes and trunks, and results are shown in the Peak Integration Results bar graph, indicating that the αA-crystallin was only detectable in eyes and not trunks and reached its highest contraction at 7 days dpf. The bar in the graph labeled Library shows the relative proportion of the y10-12 ions in the MS2 spectrum from the lens library, set at the same relative abundance as the fragment ions in 7 dpf eye digest. The relative intensities of the y10-12 fragment ions detected in the eye 4 and 7 dpf samples were very similar to those observed in the MS2 spectrum from the library, as evidenced by their dot product (dotp) values 0.98 marked above each bar.

Click here for additional data file.

Figure S6 Targeted mass spectrometric detection of αA-crystallin peptide 71–78 (FTVYLDVK) in digests of 4 and 7 day post fertilization (dpf) dissected embryo eyes and trunks

Spectral library: MS2 spectrum of αA peptide 71–78 created during a data-dependent analysis of adult zebrafish lens digest. Y5-7 fragment ions were the most abundant in this spectrum, and these were used to detect the peptide in embryo digests, based on their simultaneous elution at approximately 37.2 min during the LC/MS analyses (colored traces marked with an arrow). These fragment ion peaks were integrated for each digest from 4 and 7 dpf embryo eyes and trunks, and results are shown in the Peak Integration Results bar graph, indicating that the αA-crystallin was only detectable in eyes and not trunks and reached its highest contraction at 7 days dpf. The bar in the graph labeled Library shows the relative proportion of the y5-7 ions in the MS2 spectrum from the lens library, set at the same relative abundance as the fragment ions in the 7 dpf eye digest. The relative intensities of the y5-7 fragment ions detected in the eye 4 and 7 dpf samples were very similar to those observed in the MS2 spectrum from the library, as evidenced by their dot product (dotp) values 0.99 marked above each bar.

Click here for additional data file.

Figure S7 Targeted mass spectrometric detection of αA-crystallin peptide 89-99 (VTDDYVEIQGK) in digests of 4 and 7 day post fertilization (dpf) dissected embryo eyes and trunks

Spectral library: MS2 spectrum of αA peptide 71–78 created during a data-dependent analysis of adult zebrafish lens digest. y5, 7, and 9 fragment ions were the most abundant in this spectrum, and these were used to detect the peptide in embryo digests, based on their simultaneous elution at approximately 21.7 min during the LC/MS analyses (colored traces marked with an arrow). These fragment ion peaks were integrated for each digest from 4 and 7 dpf embryo eyes and trunks, and results are shown in the Peak Integration Results bar graph, indicating that the αA-crystallin was only detectable in eyes and not trunks and reached its highest contraction at 7 days dpf. The bar in the graph labeled Library shows the relative proportion of the y5, 7, and 9 ions in the MS2 spectrum from the lens library, set at the same relative abundance as the fragment ions in the 7 dpf eye digest. The relative intensities of the y5, 7, and 9 fragment ions detected in the eye 4 and 7 dpf samples were very similar to those observed in the MS2 spectrum from the library, as evidenced by their dot product (dotp) values ranging from 0.96–0.97 marked above each bar.

Click here for additional data file.

Figure S8 Targeted mass spectrometric detection of αBa-crystallin peptide 79–88 (HFSPDELTVK) in digests of 4 and 7 day post fertilization (dpf) dissected embryo eyes and trunks

Spectral library: MS2 spectrum of αBa peptide 79–88 created during a data-dependent analysis of adult zebrafish lens digest. b2, y8, and y9 fragment ions were the most abundant in this spectrum, and these were used to detect the peptide in embryo digests, based on their simultaneous elution at approximately 27.2 min during the LC/MS analyses (colored traces marked with an arrow). These fragment ion peaks were integrated for each digest from 4 and 7 dpf embryo eyes and trunks, and results are shown in the Peak Integration Results bar graph, indicating that the αBa-crystallin was most abundant in eyes and while detectable in 4 dpf trunks, was not observed in 7 dpf trunks. The bar in the graph labeled Library shows the relative proportion of the b2, y8, and y9 ions in the MS2 spectrum from the lens library, set at the same relative abundance as the fragment ions in the 7 dpf eye digest. The relative intensities of the b2, y8, and y9 fragment ions detected in the eye 4 and 7 dpf digests, and trunk 4 dpi digest were very similar to those observed in the MS2 spectrum from the library, as evidenced by their dot product (dotp) values ranging from 0.94–0.95 marked above each bar.

Click here for additional data file.

Table S1 Sequences of tryptic peptides used to detect proteins by mass spectrometry

The sequences of 9 tryptic peptides, 3 from each αA-, αBa-, and αBb-crytallin used to detect these proteins in digests from zebrafish embryos are given, along with their charge states, and m/z values.

Click here for additional data file.

Supplemental Information 1 Ct values for qPCR data shown in Fig. 5

Click here for additional data file.

We would like to thank Joram Piatigorsky for early conversations during the development of this project and his willingness to provide insight into lens crystallin promoter function. Jeff Gross served as a technical consultant on our work with zebrafish, Tea Meulia provided technical help with confocal microscopy (http://mcic.osu.edu/home), Jared Talbot identified muscle cell types, and Andor Kiss provided helpful feedback during the drafting of this manuscript. Ashland University undergraduate student Cassie Craig contributed to the characterization of zebrafish αB-crystallin promoters.

Additional Information and Declarations

Competing Interests

Author Contributions

Animal Ethics

Data Availability

The authors declare there are no competing interests.

Mason Posner and Larry L. David conceived and designed the experiments, performed the experiments, analyzed the data, contributed reagents/materials/analysis tools, wrote the paper, prepared figures and/or tables, reviewed drafts of the paper.

Kelly L. Murray performed the experiments, wrote the paper, reviewed drafts of the paper.

Matthew S. McDonald and Hayden Eighinger, Brandon Andrew and Zachary Haley performed the experiments, reviewed drafts of the paper.

Amy Drossman conceived and designed the experiments, performed the experiments, reviewed drafts of the paper.

Justin Nussbaum performed the experiments, analyzed the data, contributed reagents/materials/analysis tools, wrote the paper, prepared figures and/or tables, reviewed drafts of the paper.

Kirsten J. Lampi performed the experiments, contributed reagents/materials/analysis tools, wrote the paper, reviewed drafts of the paper.

The following information was supplied relating to ethical approvals (i.e., approving body and any reference numbers):

Work with vertebrate animals was approved by Ashland University’s Institutional Animal Use and Care Committee (approval number MP 2015-1).

The following information was supplied regarding data availability:

The raw qCPR Ct data is included as a Supplemental File.

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
