# Peer review of "The zebrafish as a model system for analyzing mammalian and native α-crystallin promoter function"

_PeerJ, doi:10.7717/peerj.4093_

## Round 0.1 · original submission · Major Revisions

Although the reviewers are generally positive about this study of the expression patterns of mouse and zebrafish a-crystallin promoter constructs in zebrafish, there have been problems identified with the RT-PCR data shown in Fig. 5.This is not quantitative in nature and it is unclear whether the bands are specific products. This figure should be repeated using Real-time PCR and, ideally the bands should be sequenced. Additionally, the use of at least three reference genes (e.g., Rpl13α, Ef1α, and β-actin) is recommended for the assessment of gene expression in a developmental time course in zebrafish.

Reviewer 1 ·

Basic reporting

This paper reports expression patterns of mouse and zebra fish a-crystallin promoter constructs in zebra fish embryos and examines native gene and protein expression using PCR and mass spec. There is some nice data here. The GFP reporter results are clear. They confirm that crystallin promoters can work across species, as has long been known, and also define functional fragments of the zebra fish promoters. Mass spec detected aA and aBa peptides and provided evidence for eye/non-eye regulation of aBa in development. aBb was not detected.

Unfortunately the PCR results are not so convincing. Fig 2, shows aA expression throughout development quite well. Fig 5, however, has poor image quality and, unlike tubulin, the aBa,b primers seem to be generating smaller non-specific bands. These might be solvent front, but the aA and tubulin gels are clear in this region. Were these products confirmed by sequencing? Expression in adult is obvious, but the inconsistent results between panels A and B for embryonic expression are hard to explain. Both show strong bands on single developmental days, followed by almost nothing, which by itself seems biologically unlikely. The difference in timing in different samples is also strange and looks artefactual. I think it would help if this figure was repeated using different primers. The lack of detectable peptides for aBb also raises questions about the PCR. As for the control genes, is tubulin really a good control? Might tubulin be developmentally regulated?

Although the authors cite the Hou et al 2006 paper on expression of the human bB1promoter in zebra fish in the introduction, they don’t compare their results in the Discussion. Some mention seems appropriate.

Minor points:
Line 32: Specify chaperone activity refers to a-crystallin.
Line 102: aB? Seems to have symbol character.
Line 398: Specify a-crystallin.
Figs 1,2,3: In legends – state ages of embryos for GFP expression.

Experimental design

Experimental design is fine in general, but the paper is let down by the execution of the PCR.

Validity of the findings

No further comment

Reviewer 2 ·

Basic reporting

In this study, M. Posner and coworkers conducted a series of gene reporter, expression, and proteomic studies to determine expression of zebrafish and mouse -crystallin genes in zebrafish. The A- and B-crystallins are key structural proteins of vertebrate lenses. Although some insights into transcriptional control of these genes are known for mammalian crystallins, understanding of their temporal and spatial control both in and outside of the lens requires additional studies. Zebrafish is an excellent cost effective experimental model to accelerate these studies. The findings are novel and address a number of questions in the field. The present studies thus fill a critical gap in our understanding of lens differentiation and crystalline gene expression.

The manuscript is well-written and within the scope of PeerJ. The data are of excellent technical quality. There are several minor issues to be addressed in the revision:
1) Introduction: Lens development in zebrafish and mouse differs as the mouse lens placode invaginates into the lens vesicle, prior the formation of lens fibers. Lens fiber cells in zebrafish are generated through the delamination from lens placode. This should be mentioned as both zebrafish and mouse models are discussed.
2) Methods: the coordinates of DNAs used to generate eGFP reporters should be included.
3) The paper by Zou et al. 2015, Exp. Eye Rees. 138:104-13 should be quoted here.

Experimental design

Some technical details regarding the DNAs have to be addressed (see above).

Validity of the findings

Addition of supplemental files including three video is highly valuable.

Reviewer 3 ·

Basic reporting

The authors present an excellent review of the literature. Clear and well written manuscript.

Experimental design

1. The overall purpose is unclear. Why functionally assess mouse lens promoters in zebrafish? And how would the authors determine whether negative expression in fish generated from a potentially new mouse lens enhancer would be due to species differences or merely that the enhancer was not a true enhancer in mice either?

See Intro: “This conservation suggests that mammalian a-crystallin promoters could be functionally assessed in the zebrafish, providing a faster and less expensive system than traditional mouse transgenic approaches.”

Furthermore, the authors spend a lot of effort to simply confirm that EXISTING, KNOWN regions of the mouse gene promoter function in the fish. What new is gained here, since these regions were already known to control expression in the mouse as such?
See Results: “Our results indicate that the mouse aB-crystallin promoter drives GFP expression in zebrafish embryos, and that the resulting spatial patterns reflect the functional regions first identified in mouse.”

2. The authors attempt to use promoter driven GFP to assess endogenous onset of gene expression. If an enhancer or modifier is missing in their cloning, then their entire conclusions are in error. The characterization of the developmental patterns produced by the promoters should be perform by analyzing the expression of the endogenous products.

See Results: “However, no study has characterized developmental patterns produced by the two respective promoter regions. We produced a GFP-linked 3 kb fragment of the zebrafish aBa-crystallin promoter and a series of GFP-linked fragments… We found no difference between the timing of onset for any of these zebrafish promoters… Overall these data suggest that the divergent expression of the two zebrafish aB-crystallin promoters…”

They then follow this up with RT-PCR and show that it is essentially present at all timepoints analyzed (no timepoints prior to 1 dpf were analyzed). Since this wasn’t qPCR, assessment of abundance should not be given. I am not sure that the following statement in the Discussion is valid.

See Discussion: “Lastly, by examining promoter activity, mRNA expression and protein abundance for zebrafish aBa-crystallin we resolve remaining questions about the timing of expression of this gene.”

Validity of the findings

Data are robust and controlled. See point 2 above regarding experimental design issues leading to over interpretation of findings.

Additional comments

Annotation of Figure 5 would be appreciated. Legend for colors. A-B: aA, C-D: aB.

---

## Round 0.2 · accepted · Accept

Thanks for your hard work to improve this manuscript, which adds to the body of work demonstrating why zebrafish are increasingly being utilised as a model system.

Reviewer 1 ·

Basic reporting

I think the manuscript is much improved. Indeed the authors have made several important corrections.
I only noticed two minor points:

I would drop the last sentence of the Abstract. It is speculative and doesn’t relate to these results.
Line 144 in the PDF has a square symbol character that probably should be a roman B.

Experimental design

No further comment

Validity of the findings

No further comment

Reviewer 2 ·

Basic reporting

Satisfactory.

Experimental design

Meet standards.

Validity of the findings

No comment.

Additional comments

The authors have address all main minor points raised in the earlier review.

Reviewer 3 ·

Basic reporting

No problems here. Clear and well written manuscript.

Experimental design

1. The authors have done a better job addressing the significance of the research to the field.
2. The authors attempt have resolved the issue with qPCR.

Validity of the findings

Data are robust and controlled.

Additional comments

A genuine and sincere effort was made to improve the manuscript.